# Hardware Schemes for Smarter Indoor Robotics to Prevent the Backing Crash Framework Using Field Programmable Gate Array-Based Multi-Robots

**DOI:** 10.3390/s24061724

**Published:** 2024-03-07

**Authors:** Mudasar Basha, Munuswamy Siva Kumar, Mangali Chinna Chinnaiah, Siew-Kei Lam, Thambipillai Srikanthan, Janardhan Narambhatla, Hari Krishna Dodde, Sanjay Dubey

**Affiliations:** 1Department of Electronics and Communication Engineering, Koneru Lakshmaiah Education Foundation, Green Fields, Guntur 522502, Andhra Pradesh, India; mudasar.basha@bvrit.ac.in (M.B.); msivakumar@kluniversity.in (M.S.K.); 2Department of Electronics and Communications Engineering, B. V. Raju Institute of Technology, Medak Dist., Narsapur 502313, Telangana, India; harikrishna.dodde@bvrit.ac.in (H.K.D.); sanjay.dubey@bvrit.ac.in (S.D.); 3School of Computer Science and Engineering, Nanyang Technological University, Singapore 639798, Singapore; siewkei_lam@pmail.ntu.edu.sg (S.-K.L.); astsrikan@ntu.edu.sg (T.S.); 4Department of Mechanical Engineering, Chaitanya Bharati Institute of Technology, Gandipet, Hyderabad 500075, Telangana, India; njanardhan_mech@cbit.ac.in

**Keywords:** multi-robot, backing crash prevention, sensor fusion, behavioral control

## Abstract

The use of smart indoor robotics services is gradually increasing in real-time scenarios. This paper presents a versatile approach to multi-robot backing crash prevention in indoor environments, using hardware schemes to achieve greater competence. Here, sensor fusion was initially used to analyze the state of multi-robots and their orientation within a static or dynamic scenario. The proposed novel hardware scheme-based framework integrates both static and dynamic scenarios for the execution of backing crash prevention. A round-robin (RR) scheduling algorithm was composed for the static scenario. Dynamic backing crash prevention was deployed by embedding a first come, first served (FCFS) scheduling algorithm. The behavioral control mechanism of the distributed multi-robots was integrated with FCFS and adaptive cruise control (ACC) scheduling algorithms. The integration of multiple algorithms is a challenging task for smarter indoor robotics, and the Xilinx-based partial reconfiguration method was deployed to avoid computational issues with multiple algorithms during the run-time. These methods were coded with Verilog HDL and validated using an FPGA (Zynq)-based multi-robot system.

## 1. Introduction

With the increased use of technological interventions, human needs are increasing. Service robots play a vital role in providing better services in indoor environments, and the robot population in indoor environments has gradually increased in recent years. Indoor environment constraints include limited space, and dynamics vary with respect to event conditions. A market analysis conducted by the Data Bridge market research team [1] forecasted that the global indoor robots market would be worth USD 100.37 billion by 2029, and it was valued at USD 11.65 billion at the end of 2021. This analysis requires the robot population to increase by more than 10% compared with the present situation. Policies, infrastructure, and technologies are essential to provide smooth services to humans using robots. Technological intervention is highly effective in achieving better robot services without collisions with the environment, humans, or other robots. For robots that do not collide in indoor environments, social attention [2] is a major concern. This study presents collision avoidance among multi-robots in static and dynamic scenarios.

Researchers have contributed rigorously to robotics and its applications over the past three decades. Robotics is integral to localization, mapping, path planning, navigation, sensory fusion, collision avoidance, and behavioral control algorithms, among others. Computational devices play a key role in performing these methods. Sensing is important in various applications. The development of mobile robot sensors started with tactile sensors and advanced to visual-based sensors, entailing advantages and disadvantages regarding respective sensors, as discussed in the literature. Active range sensors include LiDAR, LASAR, and ultrasonic sensors [3]. When integrating multiple sensors, sensor fusion can achieve better results with lower complexity [4]. Sensor fusion [3,5] has advanced in accomplishing a reduction in uncertainty and an increase in accuracy with reliability, as well as spatial and temporal coverage. Ciuffreda et al. [6] framed hybrid sensor fusion modeling such as PIR and ultrasonic sensors. Therefore, the proposed method was integrated with ultrasonic sensor fusion to avoid collisions. Localization methods tend to estimate robot status in the environment [7] using sensor fusion.

Most studies have focused on forward propagation during navigation and parking in indoor environments. The execution of these tasks depends on obstacles and collision avoidance. Bug and other heuristic-based obstacle avoidance methods have been deployed by researchers in the form of cooperative centralization and decentralization approaches [8,9,10]. An interesting aspect of robotics is the fact that the backing of a vehicle/robot during parking or navigation depends on its environment. Efficient solutions are required for multi-robots when their environment is cluttered with both static and dynamic objects. J. K. Suhr et al. [11] examined the challenges of a vehicle/robot backing out of a parking space using the Stixel computer vision approach. Stixel methods are used to estimate the geometry and depth of an object. When an autonomous vehicle/robot is backtracking, environmental objects must be considered, and there may be both static and dynamic objects present. Therefore, multi-robot backtracking collision avoidance systems are important for indoor service robots. Distributed robots are dependent on scheduling algorithms to avoid collisions among multi-robots. M. Asim et al. [12] investigated the scheduling algorithms in task assignment among distributed robots. Various other approaches, such as round-robin [13], auction-based [14], and first come, first served (FCFS) [15], are part of the task assignment of the scheduling approach. 

When performing collision avoidance with multi-robots, behavioral control mechanisms are essential in lane changes, whereas other methods prefer to use adaptive cruise control (ACC). ACC requires certain space policies to develop collision avoidance [16], and it consists of various features such as velocity-based, stop-and-go [17,18], and predictor-based functions [19]. The computational device has a greater impact on achieving successful implementation in a real-time scenario. Parallel computing devices are essential for multi-robot backing collision avoidance in static and dynamic scenarios. GPUs and reconfigurable computing devices provide better results than other devices. GPUs consume more power and require a larger area. Wan et al. [20] presented an FPGA-based robotic impact in real-time applications for achieving lower power consumption-based solutions and parallel computation. As mentioned by various researchers [21,22,23], FPGAs are preferable for edge computation in robotic applications. Partial reconfiguration [24,25] integration provides optimized solutions for reconfigurable edge computation and angular computations can be performed using CORDIC [26] modules for robotic applications. 

This study presents multi-robot collision avoidance in a backing scenario with the following innovations:It enables the estimation of a robot’s position in an environment using sensor fusion with hardware schemes.A hardware-based behavioral control mechanism approach is integrated with adaptive cruise control to avoid collisions.Hardware-based scheduling algorithms (round-robin (RR) and first come, first served (FCFS)) are incorporated for static and dynamic scenarios to prevent multi-robot crashes while backing up in the indoor environment.A hardware-based partial reconfiguration flow is integrated to run multiple algorithms as per the event-driven conditions.

This section presents the introduction and discusses the literature on multi-robot collision avoidance during backtracking in static and dynamic scenarios in an indoor environment. The next section elaborates on the proposed approaches that use hardware-based algorithms and their equivalent VLSI architectures. The proof validating the proposed method’s viability is presented in Section 3, with results in the form of device utilization, power consumption, experimental validation, and quantitative and qualitative comparisons with other methods. In Section 4, we conclude the article with a discussion of the merits and future scope of the proposed method.

## 2. Hardware-Based Algorithms

The proposed hardware-based algorithms are presented in this section in relation to backing crash prevention in versatile indoor parking environments in both static and dynamic scenarios. The related abbreviations are defined as listed in Table 1.

### 2.1. Hardware-Based Algorithm for Backing-Up Crash Prevention

An overview of multi-robot backing crash prevention in an indoor environment is presented in the flowchart in Figure 1a,b. The system was initiated by sensing the signals with ultrasonic sensors, which were also used for sensor fusion. Sensor fusion data have been utilized for robot localization in various parking scenarios. Parking scenarios are mostly located in an indoor environment, such as perpendicular parking with and without inclination. Robots will analyze their parking position with respect to the environment in either normal perpendicular parking or inclination parking. The next level of the proposed system estimates whether the system is in static or dynamic conditions based on the environment. The static conditions of the scheduling-based priority task assignments were utilized while traversing from the parking lot to prevent multi-robot collisions during the backing traverse with respect to the red coloured arrows as shown in Figure 1a. In a generic manner, round-robin (RR) scheduling task assignments have been incorporated for static environments. When the multi-robots are out of the parking space, fail in communication, or if there is a delay in round-robin scheduling methods, it warns the system to accommodate the dynamically related algorithm to accomplish the task. In this situation, an individual robot is fitted with first come, first served (FCFS) scheduling along with an adaptive cruise control (ACC)-based behavioral control mechanism to avoid a backing crash. The execution of the proposed methodology is described with flowchart as shown in Figure 1b.

Figure 2 illustrates the round-robin scheduling algorithm for a multi-robot system in the execution of backing crash prevention under static conditions. In this scenario, the R_2_ robot acts as the leader, and the remaining robots act as followers. Once the static conditions are confined, the leader robot takes the lead and sends a schedule to each robot. In this case, the round-robin schedule is R_3_-R_4_-R_1_-R_2_ (top to lower sequence). In this process, the leader initializes communication with the request (R) instruction signal and waits for the response from the follower robots. The latter acknowledge (Ack) their acceptance of this schedule. Once the schedule (S) assigned to the follower robot begins executing the backing up from the parking position, it communicates with the leader robot after the accomplishment (Acc) of the backing up. In this process, in the event of any communication failure among the robot’s flock, the robot aims to execute the backing up method that it prefers using the first come, first served schedule, as shown in Figure 3. In this case, R_3_ and R_1_ both opt to execute backing up simultaneously. They communicate details about their exit to all robots. However, while transitioning from the parking space, they follow adaptive cruise control (ACC) by following Equations (1) and (2).

The proposed approach followed inter-distance dynamics as per reference [27]. Specifically, we examined the dynamics that arise from the discrepancies in acceleration between leading X_L_ and following X_F_ robots. This can be effectively depicted as a double integrator system. The distance between two robots in real time is d (Equation (1)), and the reference distance is d_r_ (Equation (2)). These equations were inspired by J. J. Martinez et al. [27] regarding the convergence and stability of case 1. We considered these equations, digitized them into hardware schemes, and utilized them for collision avoidance in multi-robots.
(1)$$\iint d=\iint x_{L}-\iint x_{F},$$
(2)$$\iint d_{r}=\iint x_{L}-\iint x_{F}.$$

Velocity control is performed by a follower robot. In this approach, every robot acts as a follower and performs its action until it reaches the path. Once it attains its path, the robot follows Equations (1) and (2). 

#### 2.1.1. Hardware-Based Algorithm for Backing-Up Crash Prevention in the Static Environment

This section deals with multi-robot backing-up crash prevention in a static environment. Algorithm 1 presents the pseudocode of the robot localization in versatile parking before executing back-up traversing.
**Algorithm 1:** Pseudocode for identification of robot position during indoor parking1.  Initialize sensory distance and reference distances, digital compass directions2.      Case A: estimation of robot position in parking3.         State 1: if ((D_xθ_ = θ_ref_) && ((S_XL_ & S_XR_) = dmin))? Case B: State 2.4.         State 2: if ((D_xθ_ ≠ θ_ref_) && ((S_XL_ & S_XR_) = dmin))? Case C: Case A.5.      Case B: robot at perpendicular parking in static/dynamic state6.         State 11: if ((D_xθ_ = θ_ref_) && ((S_XB_0_ & S_XR_135_ & S_XL_225_) = dmax))? State 12: State 13.7.         State 12: Algorithm_2 of Case _A//Switch to RR & Behavioral8.         State 13: Algorithm_3 of Case _3A//Switch to FCFS & ACC algorithm9.      Case C: robot at inclination parking in static/dynamic state10.         State 21: if ((D_xθ_ ≠ θ_ref_) && ((S_XB_0_ & S_XR_135_ & S_XL_225_) = dmax))? State 22: State 23.11.         State 22: Algorithm_2 of Case _A//Switch to RR & Behavioral12.         State 23: Algorithm_3 of Case _3A//Switch to FCFS & ACC 13.  end cases.

Ultrasonic sensor data fusion with respect to the distances is memorized as per the environment, and the reference distance is the minimum (dmin) and maximum (dmax). Dx is the digital compass angles with respect to the robot alignment in the parking environment and the execution of back traversing (line 1). Algorithm 1 classifies the robot’s positioned parking with sensory information, and lines 3 to 4 present either the robot in the perpendicular or inclination type of parking. Line 3 defines robots as perpendicular parking. Perpendicular parking is localized with angles (360°/0°, 90°, 180°, and 270°), represented as θ_ref_ when S_XL_ and S_XR_ are at a minimum distance to evaluate the type of parking. Similarly, line 4 defines inclination parking when the real-time digital compass angle data reference is out of range. During perpendicular/inclination parking, when the robot wants to traverse back, it estimates that the environment contains either static objects or dynamic objects (other robots moving in parallel), as mentioned in lines 6 and 10. When robots observe the scenario, based on that sensory information, they switch to Algorithm 2 to perform back traversing in a static scenario (as mentioned in lines 7 and 11). The ellipse condition of lines 6 and 10 represents the dynamic scenario of versatile parking. As mentioned in lines 8 and 12, the robots switch to Algorithm 3 to perform back traversing under dynamic conditions.

Figure 4a,b illustrates the way in which the multi-robot system avoids collisions in static environments for both perpendicular and inclination parking. Its operational flow is presented in the form of a pseudocode in Algorithm 2.
**Algorithm 2:** Pseudocode for backing crash prevention in a static environment using RR and behavioral control algorithms1.       Initialize from Algorithm 1 and sensor fusion, robots {R_1_, R_2_, R_3_, R_4 …_ R_n_}, n = 4.2.       Case_A: Round-Robin task assignment algorithm3.       State_A1: for iteration = 1 to n, initialize the robot task assignment to backing 4.          for i = 0 to n − 1 // Iterate through tasks in a Round-Robin fashion5.                   { current robot = task assigned} 6.                if {current robot = sensory perpendicular parking}? Case_A1: Case_B.7.                if {current robot = task accomplished}? State_A1: Case_A1.8.                     i++, end case9.      Case_A1: Behavioral control algorithm@ perpendicular parking10.      State_A12: ((S_XB_0_) = dmax))? State_A13: State_A1.//Step to Backward11.      State_A13: if ((S_XB_0_ & S_XR_135_ & S_XL_225_) = dmax))? State_A31: State_A1.12.          State_A31: if ((S_XB_0_ & S_XR_135_ & S_XL_225_) = dmin))? State_A32: Backward action.13.          State_A32: if ((D_xθ_ = θ_ref_ + 90°)? State_A1: W_SR_🡪 turn @ ϑ_90_ ° & W_SL_ 🡪 Stop. 14. end case15.     Case_B: Behavioral control algorithm@ inclination parking16.      State_B1: if ((D_xθ_ ≠ θ_ref_) && ((S_XB_0_ & S_XR_135_ & S_XL_225_) = dmax))? State_B2: Case_A1.17.      State_B2: if (W_SR/L_🡪 turn @ (D_xθ_ = θ_ref_) & W_SL/R_ -> Stop)? Case_A1: State_B2.18.  end case

Algorithm 2 presents the crash prevention of autonomous multi-robots while performing backing up for versatile parking. Figure 4a shows crash prevention in a static environment during perpendicular parking. Similarly, Figure 4b illustrates the cooperative distribution. The proposed hardware-based Algorithm 2 was abstracted into three folds. Initially, multi-robots were assigned a number based on their parking space using the round-robin task assignment method (lines 2 to 5). The robots were evaluated for their localization with respect to the environment and self-analyses to determine whether they were engaged in perpendicular or inclination parking (line 6). Once the robot accomplished its task of back movement, it confirmed its status to the team leader (line 7). The multi-robot backing-up crash prevention of the behavioral control algorithm for perpendicular parking is defined in lines 9–14. The current robot initializes to evaluate its distance using the backward sensor; when the objects/robots are not available in their maximum distance radius, it initializes the backward action (line 10). It confines the sensory information of the left and right sensors with respect to their angles to avoid collisions (line 11). The current robot executes the back-up movement until it achieves its positional distance from its parking space and maintains the minimum distance from other objects (line 12). Once it obtains sufficient space to turn 90° to its left, the gateway spotted on the left takes the opposite direction (line 13). It confirms all its actions and accomplishments with the team’s master robot. Similarly, when robots are engaged in inclination parking, backing crash prevention is achieved, as presented in Algorithm 2 in lines 15–18. In this scenario, robots are wisely dependent on ultrasonic sensory fusion and digital compasses. The digital compass angular information deviates with respect to perpendicular parking as the reference angle as one condition of line 16. The other condition is the free space for backpropagation with ultrasonic sensors. It performs angular correction of the robot with respect to the environment and continues to perform the backward action until it positions itself in relation to the reference line of the environment or the minimum distance of the other objects (line 17). 

#### 2.1.2. Hardware-Based Algorithm for Backing-Up Crash Prevention in the Dynamic Environment

Figure 5a,b shows the backing-up crash prevention of the multi-robots in the dynamic conditions of versatile environments, such as perpendicular and inclination parking. Algorithm 3 presents the pseudocode of multi-robot backing-up crash prevention in dynamic, versatile environments.
**Algorithm 3:** Pseudocode for backing-up crash prevention in a dynamic environment using FCFS and ACC algorithms1.       Initialize from Algorithm 1 and sensor fusion, robots {R_1_, R_2_, R_3_, R_4_ … R_n_}, n = 4.2.       Case 3A: 3.       State_3A1: if ((D_xθ_ ≠ θ_ref_) && ((S_XB_0_ & S_XR_135_ & S_XL_225_) = dmax))?4.                          {Algorithm 2_ Case _B}: {Algorithm 2_Case A}5.                         else6.                          { State_3A2}7.       State_3A2: if (diff {D_SF___B_ (@t), D_SF___B_(@t−1)}> D_SF___CON_)? Case 3B: Algorithm_2.8.      end case9.      Case 3B: FCFS & ACC approach10.       State_3B1: if (D_SF___B_ ≠ D_SF___CON_)? State_3B2: Algorithm_2.11.       State_3B2: if (R_n_ = R_odd_)? State_3B3: State_3B4.12.       State_3B3: R_odd_🡪{Left turn (θ_45°), Odometer_hyp, Right turn (θ_135°)} 13.       State_3B4: R_even_🡪{Right turn (θ_45°), Odometer_hyp, Left turn (θ_135°)}14.  end case

Under dynamic conditions, where each individual robot has distinct prior tasks to fulfill, all the robots must kick start from parking. In such a scenario, the first come, first served (FCFS) method and adaptive cruise control (ACC) integration provide the best solution under dynamic conditions. At every moment, the algorithm makes a decision based on the movement of the robots, and while approaching the objects, the scenario is static or dynamic based on the sensory fusion of the ultrasonic sensors (lines 2 to 8). The dynamic condition is defined based on the difference in velocity and distance between the current robot and other robots/objects in the environment (line 7). The dynamic behavioral control approach is mentioned in lines 9–14. According to this approach, robots are classified as odd (R_1_, R_3_) and even-type (R_2_, R_4_) robots. The robots positioned on the left side of the plane are assigned odd numbers, and the other side is assigned even numbers (line 10) and their back traversing of all robots have been represented with red coloured arrows as shown in Figure 5a(A–F),b(A–F).

Before starting the robot’s validation, a speed of 0.25 m/s (D_SF_CON_) was considered to be a constant speed for each robot. The real-time distance difference with respect to the time interval was evaluated and compared in the environment (line 11). FCFS is an inherent parameter of the computation, as the current robot takes the opportunity to execute its action without colliding with other robots using ACC. Among the various types of ACC, we incorporated the stop-and-go approach for this method. Figure 5c presents the action movements of the Algorithm 3 pseudocode in lines 12 and 13. As illustrated in Figure 5c, odd robots perform a left-turn action with an angle of 45°, and even robots perform a right-turn action with an angle of 45°, as shown in Figure 5c(A) with the representation of dotted lines. The multi-robot system traverses the hypothesis equivalent distance using the internal counter-based odometer method, as illustrated in Figure 5c(B) with the representation of dotted lines. The inverse turn to the previous angular action is completed by an odd robot with a right turn of 135° and a robot with a left turn of 135° (lines 12–13), as illustrated in Figure 5c(C) with the representation of dotted lines. 

### 2.2. Hardware Schemes for Backing-Up Crash Prevention

Backing-up crash prevention methods have been addressed using hardware schemes that are amenable to real-time implementation. Figure 6 presents an overview of the proposed multi-robot backing-up crash prevention approach. 

The environment was sensitized using ultrasonic sensors with a digital compass, which was communicated between the multi-robots using the ESP8266 Wi-Fi module manufactured by Espressif Systems in Shanghai, China and interfaced with a UART 32-bit module operated at 9600 baud rates. Ultrasonic sensors were initiated using a control unit every 1/3 s, and echo signals were digitized into a distance using the pulse–width modulation technique. The ultrasonic sensor distances stored in the 32 × 108 (width and depth) FIFO module and AXI-based FIFO control were integrated into the sensor fusion module, as shown in Figure 6. The 32-bit data are driven out of the module, which is integrated with AXI-based FIFO control data of 12 bits and appended with the original 20-bit distance. The control data of 12 bits define the respective sensor and its position on the robot. The proposed approach was integrated for both static and dynamic scenarios. In the static scenario, multi-robots prefer round-robin-type task assignments while performing backing-up from the parking space. Similarly, while performing in dynamic scenarios, FCFS and ACC methods have guided multi-robots to avoid crashes between other robots and the environment. The entire system is operated under event-based conditions using the control unit, and it synchronizes various frequencies and interfaces. The novelty of the proposed method was achieved with the effective utilization of partial reconfiguration tools of Xilinx to decrease power consumption and obtain effective synchronization. The executive module was embedded with a motor control for both the stepper and servomotors.

#### 2.2.1. Hardware Schemes for Multi-Robot Backing-Up Crash Prevention in a Static Scenario

Figure 7 presents backing-up crash prevention in the static scenario, which is embedded with the behavioral decision module, the round-robin (RR) switching network module, and the static backing crash module. The behavioral decision module is part of the partial reconfiguration module, which determines the selection of static and dynamic approaches. The robot angular values were evaluated using a digital compass with respect to onboard real-time, and the Xilinx CORDIC IP core (acts as the reference angle) was utilized for next-level digitization. The behavioral decision module estimates the localization of the robot in either perpendicular or inclination parking under static and dynamic conditions. For inclination parking, it corrects its angular position into a straight-line approach using real-time digital compass data and CORDIC IP cores. 

One of the novel approaches for multi-robot backing-up crash prevention is to use the RR module for task assignment among FPGA-based multi-robots. The RR switching network fetches the input data from Wi-Fi, and among the multi-robots, R_2_ acts as the leader. It assigns the target to each robot and starts the sequence as R_3_, R_4_, R_1_, and R_2_. After backing-up is carried out by an individual robot, the leader robot communicates its accomplishments. The leader robot performs backing-up in the last robot sequence. Under any conditions, the robots are unable to perform a backing-up crash, and the sequence is changed by the leader. In this regard, the arbiter role (AR) with the highest priority is classed as AR1 and the lowest priority as AR4. The static backing crash prevention module operates using the RR switching module, the decision module, and sensory fusion. It makes decisions based on the event conditions and performs backing-up of the multi-robots in sequence without collisions. It performs actions such as backing-up and turning angular movement of 90° with respect to the hardware scheme algorithms. 

#### 2.2.2. Backing-Up Crash Prevention in Dynamic Scenarios Using Hardware Schemes for Multi-Robots

Backing-up crash prevention in dynamic scenario-related internal hardware schemes is illustrated in Figure 8. Regular task assignments are not feasible in emergency conditions; in this regard, multi-robots that play roles individually at the same time crash into other robots. To avoid such a scenario, the proposed approach avoids collisions by integrating the FCFS and ACC modules. PR flow enables this to be a dynamic scenario. FCFS is embedded in a velocity evaluation module (VEM) and position reference_FIFO module. The VEM continuously provides the difference between the present and past distances of the robot, and it accommodates the velocity of other objects. In parallel, while the robot is in a backing position, its position is self-analyzed and registered in the FIFO. Based on this sensory and FCFS decision, ACC performs the stop-and-go method. ACC is performed based on the robots, which are classified as odd and even numbers according to their localization in the environment, as shown in Figure 5c.

ACC stops an individual robot until it observes that another robot has cleared the space required to execute its kinematics. The go operation of the ACC is performed using three counter designs: clockwise turn, traversing distance of the hypothesis based on its localization, and anti-clockwise turn towards the gateway.

## 3. Results

The proposed multi-robot backing-up crash prevention-related results are addressed in this section in the form of a multi-robot setup, hardware algorithm resource utilization, and its power analysis with experimental validation. The FPGA-based accelerators, as shown in Figure 6, Figure 7 and Figure 8, were developed with Verilog HDL, simulated and synthesized with Xilinx tools Vivado 2017.3, and licensed under Xilinx University. They were deployed using a field-programmable gate array (FPGA) device, which was utilized for validation of the proposed multi-robot backing-up crash prevention. 

### 3.1. Resource Utilization

Our state-of-the-art approach is a hardware scheme for multi-robot backing-up crash prevention with dynamic partial reconfiguration. In this study, after digitization from HDL to bit stream generation with the Vivado tool, the bit streams were re-stored in a 4 GB SD card. Based on the event condition, the bit files were retrieved through the AXI lite and the processing system (PS) of the Zed board. While conducting the experiment in real-time, the fetch bit stream was operated using the programmable logic (PL) of the Zed board. This switching was performed using a dynamic partial reconfiguration decision control module. PL was operated with a clock frequency of 100 MHz, interfaced with various other frequency sensors, and synchronized with AXI lite.

An XC7Z020 Zed board was manufactured by Xilinx, San Jose, CA, USA. Its programmable logic cells are around 85 K, and its overall device is integrated with look-up tables (LUTs) and flip flops, which are used for logic operations and short memory registrations. BRAM’s operation through AXI lite consists of 36 kb of each block, over 140 blocks (4.9 Mb), utilized for the storage of the sensor fusion data and intermediate data storage. In the proposed design, most of the FIFOs utilized BRAM. Other computations, such as data transfer and computations, were performed using DSP slices; there are approximately 220 (18 × 25 MACCs). These resources were used to execute the proposed approach, as detailed in Table 2.

Zed board FPGA resources are fewer among the available FPGA boards. Researchers have attempted to optimize the approach to achieve cost-effective results. In this regard, the experimentation was conducted in two flows: general implementation and PR flow. The general flow consumed approximately 82% (43,506), 77% (108), and 64% (140) of the LUTs, BRAM, and DSP slices, respectively, as shown in Table 2. Moreover, the static power consumption was around 2.4 watts with respect to device utilization. According to one study [23], as device consumption increases, it affects the overall performance and power, and it also generates operation glitches. At the same time, considering the trade-off between cost and technology, the proposed method addressed backing-up crash prevention using the PR flow approach.

Figure 9 presents a quantitative comparison of device utilization with respect to the general flow versus PR flow. Xilinx-based ILA was used as a monitoring tool for evaluating devices that utilize individuals while executing the PR flow. The LUTs, BRAM, and DSP slices are addressed for the following modules of backing crash prevention with respect to static at perpendicular as 40%, 46%, and 46%; static at inclination as 43%, 47%, and 47%; dynamic at perpendicular as 46%, 51%, and 47%; dynamic inclination as 48%, 53%, and 48%, respectively. Similarly, the static power consumption of the above modules, monitored with the Xilinx power estimator (XPE) illustrated in Figure 10, is 1.18 watts, 1.26 watts, 1.34 watts, and 1.4 watts. 

### 3.2. Experimental Results

This section deals with the validation of the proposed research work in the form of experiments. In the validation process, we developed test beds in the form of mobile robots. 

#### 3.2.1. Experimental Setup 

Mobile robots were fitted with mechanical, electrical, and computational devices. Four pairs of ultrasonic sensors were positioned in the four directions of each robot and spaced every 90°. The left and right sides of each robot were deployed on a servo motor. Each robot was powered using 24 volt and 7 amp lead acid batteries. The battery voltages were downsampled to 5 V using the voltage regulator module 7805 and supplied as a source for the electronic components and computational devices. Stepper motors were positioned on both sides of each robot’s frame. For the frame, the bottom layer was embedded with stepper motors, the top layer positioned with batteries, and the next layer positioned with electronic components and computational devices. The top layer’s sides were interfaced with the sensors and digital compass. The complete experimental setup of the mobile robot is shown in Figure 11a. The environment of the experimental setup is shown in Figure 11b.

#### 3.2.2. Experimental Results of Multi-Robot Backing-Up Crash Prevention in a Static Scenario

Figure 12a–f shows the experimental validation of backing-up crash prevention in a static scenario where robots were engaged in perpendicular parking. The proposed hardware-based algorithms and their architecture are shown in Figure 4a, and five experimental validations are presented in Figure 12a–f. According to the proposed algorithm, leader robot R_2_ was positioned at the even spot (left side and other end of the environment), as shown in Figure 12a–f. It took the lead, communicated with the robots through Wi-Fi, and used the RR task assignment method. For example, R_3_, which was driven out of the environment, is presented in Figure 12a. As per the RR method, it encouraged the next robot, R_4_, to drive backwards without colliding with the environment and other robots, as shown in Figure 12b. In RR, the next robot was R_1_, which exited the parking space, as illustrated in Figure 12c. After R_1_ had accomplished its task, the leader robot, R_2_, drove backwards and joined the team as per the application of the next task. This flow is illustrated step-wise in Figure 12d–f. The experimental demonstration was posted on M.C.C.’s YouTube channel: https://www.youtube.com/watch?v=bx0eihntOlo (accessed on 18 January 2024). Similarly, when robots were in an inclined parking space, they initially corrected their positions and executed actions in a line formation, as shown in Figure 12a–f.

#### 3.2.3. Experimental Results of Multi-Robot Backing-Up Crash Prevention in a Dynamic Scenario

Dynamic backing-up crash prevention was validated through the experimental results, as illustrated in Figure 13a–h. The dynamic backing crash was accomplished using the hardware schemes of the FCFS and ACC, as shown in Figure 8. In this experiment, all robots were trying to exit parking spaces. The FCFS method was applied to each robot, its position in the environment was identified, and the ACC rules were applied. Once initialized, R_3_ took its position as the lead between R_4_ and R_3_. R_4_ applied the ACC rule as stop-and-go, as illustrated in Figure 13a. While R_4_ was prepared for its position and movement, R_1_ intermittently took the environmental lead and accomplished its backing crash prevention task. Once again, R_4_ was waited for until R_1_ accomplished its task using the ACC rule. R_2_ was completely blocked by all robots in accomplishing the exit from the parking space using dynamic backing-up crash prevention. In this regard, R_2_ exited last and joined the group as per the next level application, and Figure 13a–h presents the same flow. The same experimental demonstration was posted on M.C.C.’s YouTube channel: https://www.youtube.com/watch?v=T_e9dIBJHi0 (accessed on 18 January 2024). Similarly, when robots were in an inclined parking space, they initially corrected their positions and executed actions in a line formation, as shown in Figure 13a–f. 

Table 3 presents the relevant fields of backing-up crash prevention methods. Several studies [11,28,29] have used a camera to estimate rearview objects. Automation extensions were observed in [11]. However, the camera’s computation and power consumption are high, and fusion between images is not available. Other researchers [28] evaluated pedestrian backing crash prevention using a warning system. Another study [30] evaluated dynamic obstacles and avoided the use of differential-driven wheeled mobile robots. Overall, in this comparison and literature survey, as per reference [20], few researchers have contributed towards obstacle avoidance using FPGA-based edge computation solutions for robotics. In this regard, the proposed method provides a solution for multi-robot backing crash prevention using the RR, FCFS, and ACC methods with PR flow. 

## 4. Conclusions

In this study, the research contributions are the algorithms for the prevention of backing-up crashes of multi-robots using novel hardware schemes and partial reconfiguration (PR) methods. The state of the art in this research work is as follows: estimation of the multi-robot positions in versatile environments, such as perpendicular and inclination parking. Hardware schemes for scheduling methods, such as the round-robin (RR) method, have been used for communicating to exit from parking spaces by multi-robots to prevent backing-up crashes in static scenarios. The behavioral control mechanism was integrated with the first come, first served (FCFS) and adaptive cruise control (ACC) algorithms to enable performance in dynamic scenarios. This was coded using Verilog HDL and tools with Xilinx Vivado, and the schemes were deployed in multi-robots embedded with a Zed board FPGA as an edge computational device. The system was validated in two ways: general flow and PR flow. The device utilization was very high in general flow: 82% of look-up tables (LUTs), 77% of Block RAM (BRAM), and 64% of DSP Slices were occupied. In this regard, the PR flow provided optimized device utilization of 40–48% of LUTs, 46–53% of BRAM, and 46–48% of DSP slices for both static and dynamic scenarios. Proportionally, the device utilization affected the static power consumption of the general flow, which was 2.4 watts, and the PR flow, which ranged from 1.26 to 1.4 watts. The future scope of this work is the implementation of backing-up crash prevention for n robots in indoor environments.

## Figures and Tables

**Figure 1 sensors-24-01724-f001:**
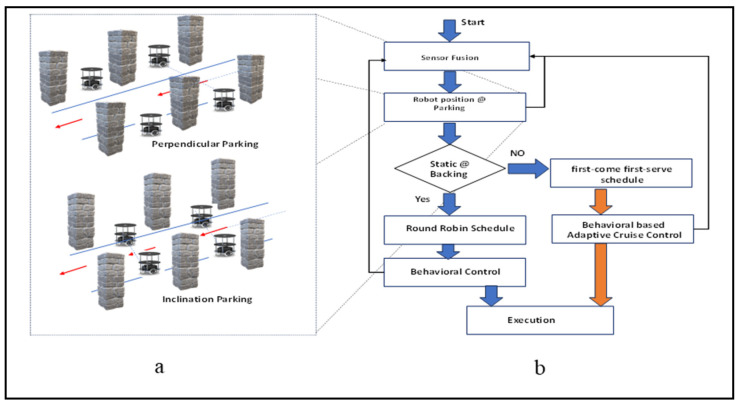
(**a**,**b**) Flowchart of multi-robot backing crash prevention in indoor environments.

**Figure 2 sensors-24-01724-f002:**
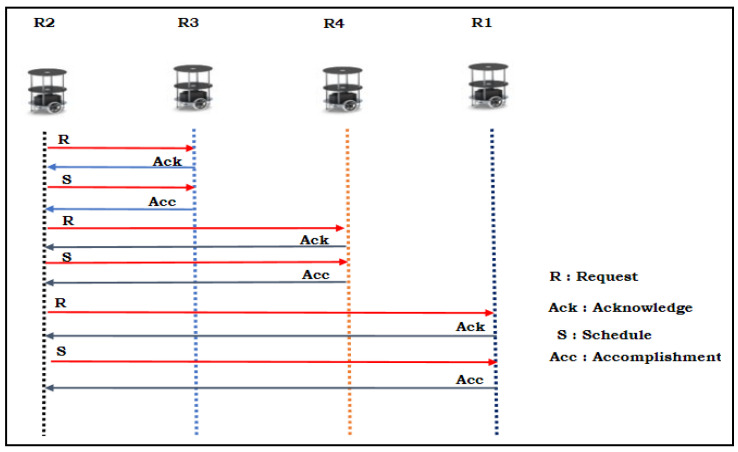
Round-robin scheduling between the multi-robots for backing crash prevention in static conditions.

**Figure 3 sensors-24-01724-f003:**
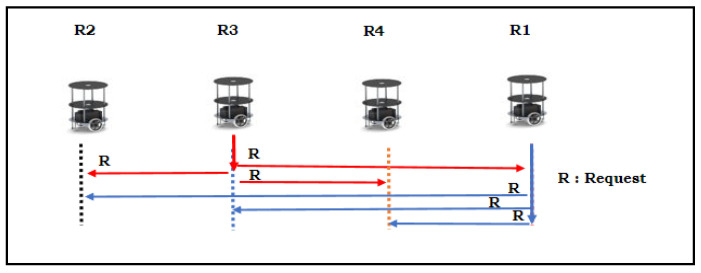
First come, first served scheduling between the multi-robots for backing crash prevention in dynamic conditions.

**Figure 4 sensors-24-01724-f004:**
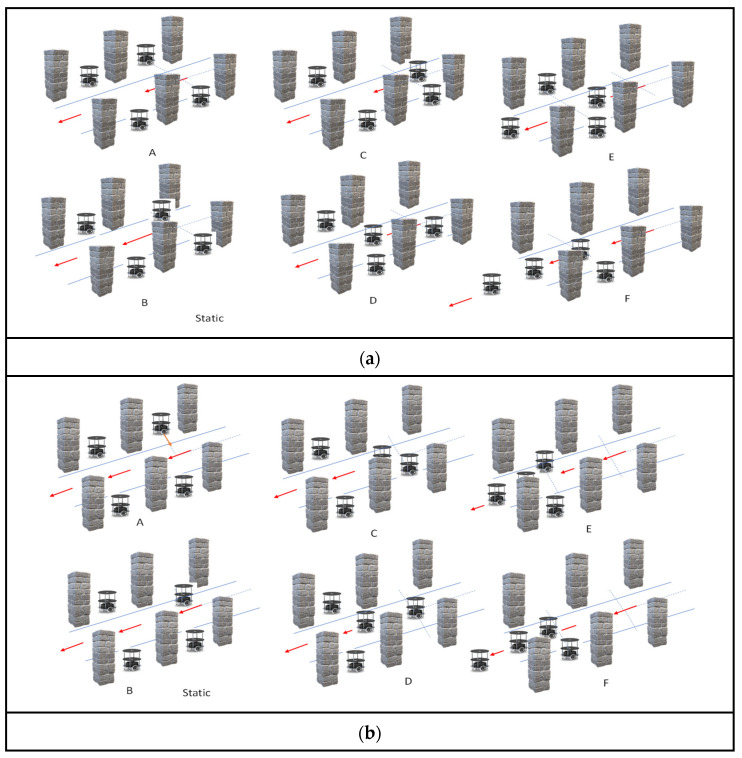
(**a**) Multi-robot backing-up crash prevention in perpendicular parking in a static environment. (**b**) Multi-robot backing-up crash prevention in inclination parking in a static environment.

**Figure 5 sensors-24-01724-f005:**
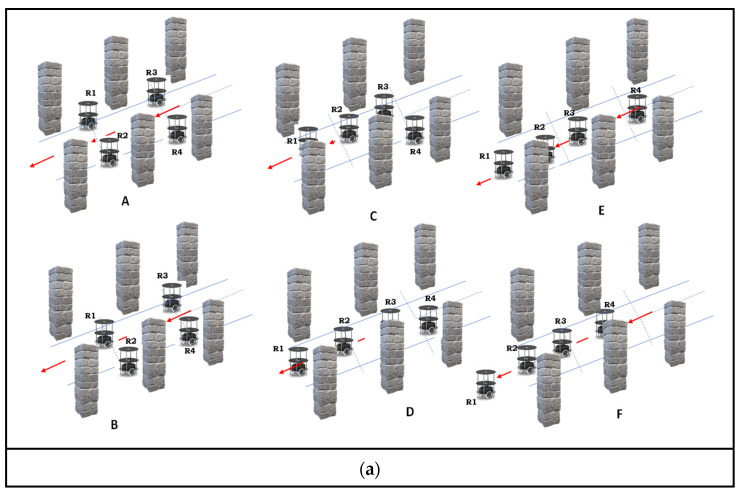
(**a**) Multi-robot backing-up crash prevention during perpendicular parking in a dynamic environment. (**b**) Multi-robot backing-up crash prevention during inclination parking in a dynamic environment. (**c**) Multi-robot adaptive cruise control for backing-up crash prevention in a dynamic environment.

**Figure 6 sensors-24-01724-f006:**
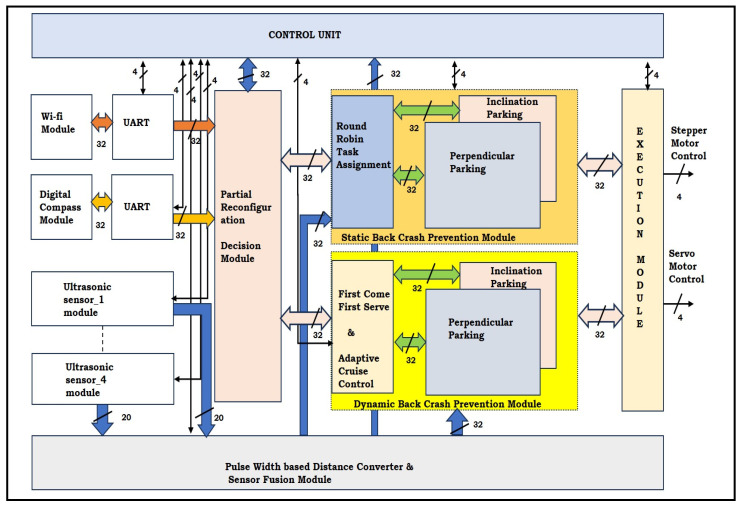
Overall hardware scheme for multi-robot backing-up crash prevention.

**Figure 7 sensors-24-01724-f007:**
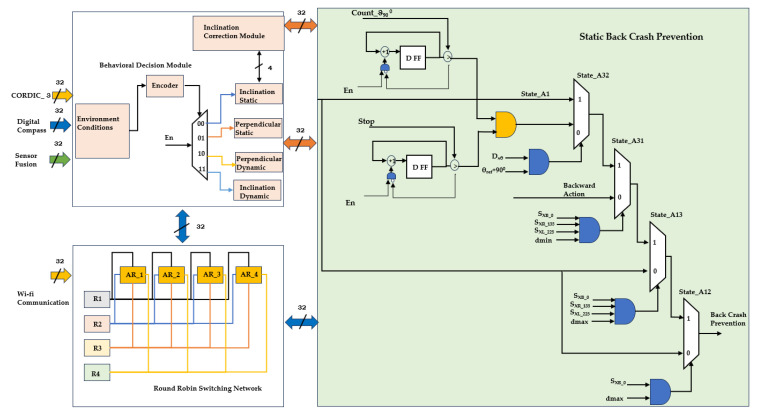
Internal architecture of multi-robot backing-up crash prevention in a static scenario.

**Figure 8 sensors-24-01724-f008:**
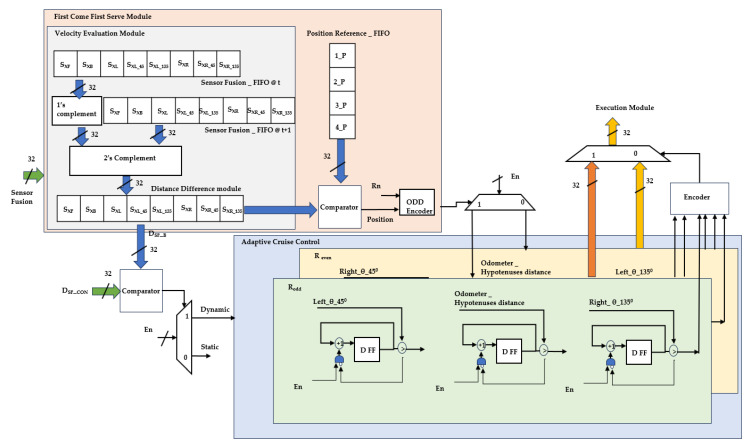
Hardware scheme for multi-robot backing-up crash prevention in a dynamic scenario.

**Figure 9 sensors-24-01724-f009:**
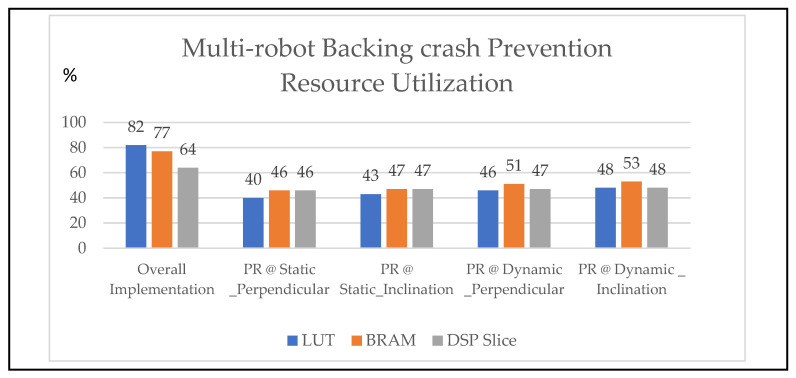
Backing-up crash prevention resource utilization of multi-robots using general and PR flow.

**Figure 10 sensors-24-01724-f010:**
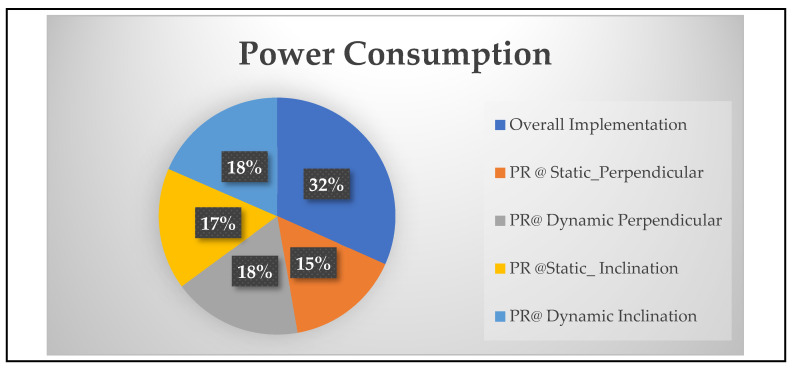
Device power consumption for backing-up crash prevention in a versatile environment using PR Flow.

**Figure 11 sensors-24-01724-f011:**
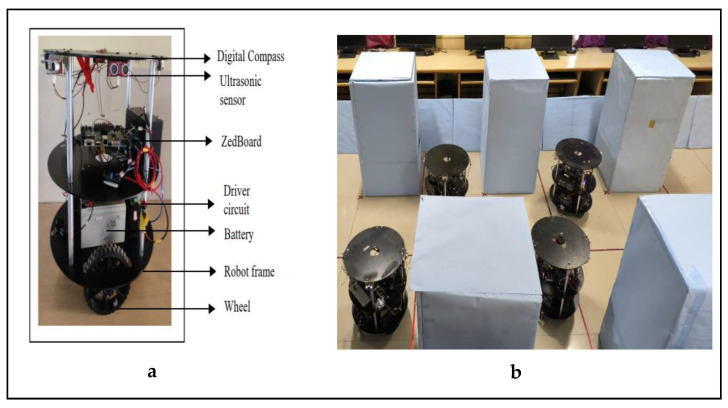
(**a**,**b**) Experimental setup of mobile robot.

**Figure 12 sensors-24-01724-f012:**
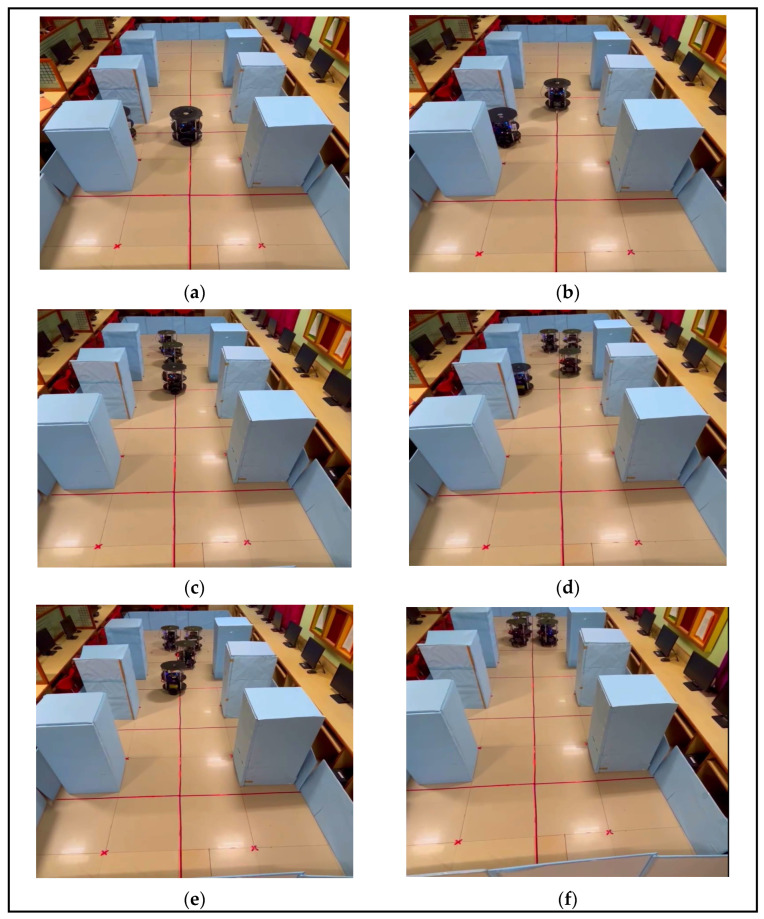
(**a**–**f**) Experimental results of static backing-up crash prevention with multi-robots.

**Figure 13 sensors-24-01724-f013:**
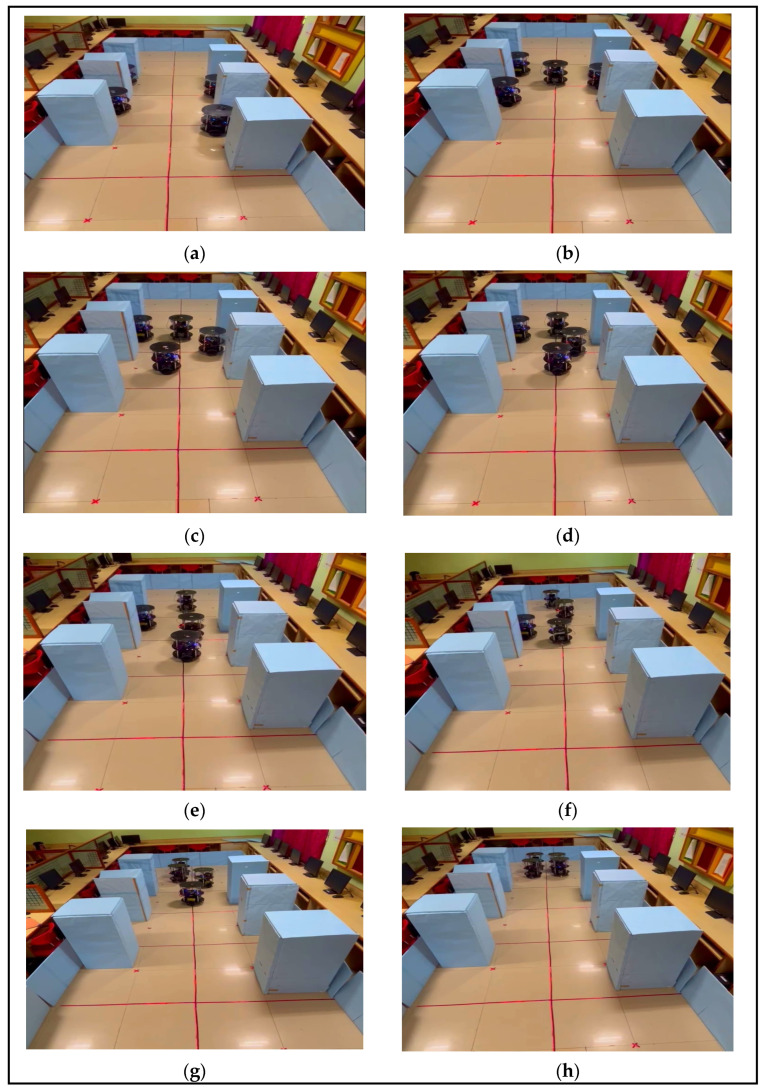
(**a**–**h**) Experimental results of dynamic backing-up crash prevention with multi-robots.

**Table 1 sensors-24-01724-t001:** Proposed research-related abbreviations.

Symbol	Abbreviation
S_XY_	S: Ultrasonic sensor. X: Represented robot type as R_1_…Rn. Y: Represented at F: front, B: back, L: left side, R: right side.
M_SY_	M: Motor. S: Servo motor. Y: Motor positioned at F: front, B: back, L: left side, R: right side.
W_ST_	W: Wheel connected to motors. S: Stepper motor. T: Right or left.
D_SF_X_	D_SF_: Sensor fusion distance data, X: sensor position on robot.
D_xθ_	Digital compass of robots.
θ_ref_	Reference angles as 90°, 180°, 270°, and 0° or 360°.
R_XV_	R: Robot, X: assigned robot number, V: velocity.

**Table 2 sensors-24-01724-t002:** Zed board FPGA resource utilization for backing crash prevention.

Module	LUT	BRAM	DSP Slice
Interfacing modules (sensors, motors, Communication (UART), Xilinx IP cores)	6852	24	36
Static backing-up crash prevention @ perpendicular	4788	8	10
Static backing-up crash prevention @ Inclination	5852	10	12
Control unit and PWDC sensor fusion	4468	20	42
Partial Reconfiguration module	5586	12	14
Dynamic backing-up crash prevention @ perpendicular	7448	16	12
Dynamic backing-up crash prevention @ Inclination	8512	18	14
Total	43,506	108	140

**Table 3 sensors-24-01724-t003:** Comparison of multi-robot backing-up crash prevention with relevant research methods.

Reference Works	Sensory Approach		Algorithm	Hardware	Advantages	Disadvantages
	Method	Fusion				
[11]	RCB-D camera	X	Stixel generation	CPU	Spare 3D points spread in a wide field of view	High computational challenges
[28]	RCB-D camera	X	Pose-specific pedestrian recognitions	CPU	Pedestrian detection in rear view	Limited with warning
[29]	LiDAR and stereo camera	X	Exploiting planar edge point to back-projected plane geometric constraints	CPU	Comparison between LiDAR and camera	Computation and power consumption
[30]	-	X	Dynamic obstacle avoidance of differential-drive wheeled mobile robot	CPU	Skidding and slipping analysis in obstacle avoidance	Limited to simulation
Proposed	Ultrasonic sensor	√	Backing-up crash prevention for both static and dynamic scenarios	FPGA	Partial Reconfiguration-based hardware schemes are a novel approach	Geometry-based analysis will be incorporated in future

X—Fusion is not available, √—Fusion is available.

## Data Availability

Data are available for experimental validations in both static and dynamic environments. Static—https://www.youtube.com/watch?v=bx0eihntOlo, dynamic—https://www.youtube.com/watch?v=T_e9dIBJHi0 (accessed on 18 January 2024).

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
