# Peer review of "Hardware Schemes for Smarter Indoor Robotics to Prevent the Backing Crash Framework Using Field Programmable Gate Array-Based Multi-Robots"

_sensors, 2024, doi:10.3390/s24061724_

Round 1
Reviewer 1 Report
Comments and Suggestions for Authors
The article presents a hardware architecture based on FPGA for the control and collision prevention of robots.
The article is well written and describes correctly the architecture and the tests, and it is supported by a couple of videos to show the results obtained.
I have a few comments on the article which I list below.
1 - The figure showing the movement of the robot should be improved to better clarify the operation (Fig. 1, 2.x and 3.x).
2 - Parts of fig5 and fig6 are difficult to read.
3 - The ZedBoard device is written together, instead of Zed board.
4 - In section 2 the robot is presented, but if I understand correctly, only the ultrasonic sensors are used in the algorithm, however, there is a sensory fusion stage. I think this should be clarified in the text and which parts of the robot will be used for the proposal.
Comments on the Quality of English LanguageThe articule is good, I think that minor english edition is necesary.
Author Response
Dear Reviewer, Thank you for providing the corrections and suggestion. Kindly PFA of the responses for your Comments.

Reviewer 2 Report
Comments and Suggestions for Authors
1. Please remark the novelty in the Abstract.
2. Although the document explains well the development of the project, the main base is certainly not described. I repeat here part of the Abstract:
"The behavioral control mechanism of the distributed multi-robot was integrated with scheduling and Adaptive Cruise Control (ACC) algorithms. The integration of multiple algorithms is a challenging task for smarter indoor robotics, and the Xilinx-based partial reconfiguration method has been deployed to avoid computation issues of multiple algorithms at runtime".
Neither scheduling nor Adaptive Cruise Control (ACC) are explained here. No equations, no convergence/stability analysis is provided.
The inclusion of many algorithms should be (briefly) explained from the Abstract and described in detail in the document.
Besides, an important issue is the estimation part and this topic is also missing. How/with which structure, equations is this achieved? How do you guarantee convergence and stability again? No clues.
Please explain well these parts.
Comments on the Quality of English Languageok
Author Response

(The authors gave the same response as above.)

Round 2
Reviewer 2 Report
Comments and Suggestions for Authors
During the 1st review, I numerated the issues that needed attention.
Please add a check-list to follow your modifications.
Comments on the Quality of English Language
No big problem.
Author Response
Dear Reviewer,
we have provided responses for your comments. Kindly Please find attachment
